# Phase I trial of WEE1 inhibition with chemotherapy and radiotherapy as adjuvant treatment, and a window of opportunity trial with cisplatin in patients with head and neck cancer: the WISTERIA trial protocol

Anthony Kong,[1] James Good,[2] Amanda Kirkham,[3] Joshua Savage [ID],[3] Rhys Mant,[3] Laura Llewellyn,[4] Joanna Parish,[1] Rachel Spruce,[1] Martin Forster,[5] Stefano Schipani,[6] Kevin Harrington,[7] Joseph Sacco,[8] Patrick Murray,[9] Gary Middleton,[10] Christina Yap [ID],[3] Hisham Mehanna [ID][1]

AKo and JG contributed equally.

CY and HM are joint senior authors.

**Correspondence to**
Professor Hisham Mehanna;
h.mehanna@bham.ac.uk

## ABSTRACT

**Introduction** Patients with head and neck squamous cell carcinoma with locally advanced disease often require multimodality treatment with surgery, radiotherapy and/or chemotherapy. Adjuvant radiotherapy with concurrent chemotherapy is offered to patients with high-risk pathological features postsurgery. While cure rates are improved, overall survival remains suboptimal and treatment has a significant negative impact on quality of life.

Cell cycle checkpoint kinase inhibition is a promising method to selectively potentiate the therapeutic effects of chemoradiation. Our hypothesis is that combining chemoradiation with a WEE1 inhibitor will affect the biological response to DNA damage caused by cisplatin and radiation, thereby enhancing clinical outcomes, without increased toxicity. This trial explores the associated effect of WEE1 kinase inhibitor adavosertib (AZD1775).

**Methods and analysis** This phase I dose-finding, open-label, multicentre trial aims to determine the highest safe dose of AZD1775 in combination with cisplatin chemotherapy preoperatively (group A) as a window of opportunity trial, and in combination with postoperative cisplatin-based chemoradiation (group B).

Modified time-to-event continual reassessment method will determine the recommended dose, recruiting up to 21 patients per group. Primary outcomes are recommended doses with predefined target dose-limiting toxicity probabilities of 25% monitored up to 42 days (group A), and 30% monitored up to 12 weeks (group B). Secondary outcomes are disease-free survival times (groups A and B). Exploratory objectives are evaluation of pharmacodynamic (PD) effects, identification and correlation of potential biomarkers with PD markers of DNA damage, determine rate of resection status and surgical complications for group A; and quality of life in group B.

**Ethics and dissemination** Research Ethics Committee, Edgbaston, West Midlands (REC reference 16/WM/0501)

## Strengths and limitations of this study

► The study is one of the first to explore a WEE1 inhibitor in combination with cisplatin and radiotherapy, and the first to do so in head and neck cancer.

► Allows collection of tissues for translational research to examine the pharmacodynamic effects of WEE1 and WEE1 in combination with cisplatin.

► Uses an efficient time-to-event continual reassessment method design which allows dose assignments to be performed with the flexibility of continual patient accrual or temporary accrual suspension to permit sufficient accumulation of safety information as necessary.

► Additional biopsies in the preoperative setting may affect the patient enrolment.

► The oral administration of AZD1775 may affect recruitment in postoperative patients if they have difficulty or are unable to swallow.

initial approval received on 18/01/2017. Results will be disseminated via peer-reviewed publication and presentation at international conferences.
**Trial registration number** ISRCTN76291951 and NCT03028766.

## INTRODUCTION

Head and neck cancers are the sixth most common cancer worldwide,[1] with 12 000 new cases per year in the UK (Cancer Research UK (CRUK) head and neck cancer statistics). The most common types are head and neck squamous cell carcinom (HNSCC) arising from laryngeal, oral cavity, oropharynx and hypopharyngeal subsites.[2]

Primary surgery is the standard of care for cancers of the oral cavity, and for more advanced cancers of the larynx and hypopharynx, while chemoradiation is the standard approach to oropharyngeal disease. Postoperative radiotherapy (PORT) or postoperative chemoradiation (POCRT) is often recommended in the setting of high-risk features such as a positive margin, multiple involved nodes, extracapsular spread and perineural infiltration.

The role of concomitant chemotherapy was established by the Meta-Analysis of Chemotherapy on Head and Neck Cancer in 2009.[2] This study pooled individual patient data from 93 trials involving 16 485 patients receiving either definitive local therapy alone (surgery or radiotherapy) or local therapy plus chemotherapy (induction, concomitant or adjuvant). Concomitant therapy was assessed in 50 trials involving 9605 patients; chemotherapy decreased the risk of death with an HR of 0.81, translating into a 6.5% overall survival advantage at 5 years as a result of improved disease-free survival. No benefit was seen in patients aged over 70 years. This provides robust evidence that the ability of radiotherapy to secure local control can be enhanced by systemic therapy.

Acute toxicity and long-term sequelae of POCRT have a considerable negative health impact, and despite intensive treatment, 5-year overall survival is suboptimal at 53%.[3] Locoregional relapse is particularly difficult to salvage, and local control is, therefore, closely correlated with overall survival. Morbidity imposed by local recurrence in the absence of distant metastatic disease is considerable and can be extremely challenging to palliate adequately. There is an urgent need to develop novel approaches that achieve improved locoregional disease control and reduce treatment-related morbidity for this patient group. Achieving this may translate into improved overall survival and an enhancement in patient-related outcome measures.

## Chemotherapy, radiotherapy and DNA damage response

Cellular DNA damage response (DDR) is central to the preservation of genomic integrity. Cancer development is a multistep process where deregulation of the DDR contributes to phenotypes such as sustained cell proliferation and resistance to cell death. This increase in endogenous DNA damage necessitates the acquisition of compensatory mechanisms if the cell is to avoid death through uncontrolled genomic instability. Importantly, these compensatory changes may constitute a molecular 'Achilles heel' that is vulnerable to therapeutic exploitation with a new generation of targeted agents. POCRT exploits the differential DDR in malignant and normal tissues to eradicate microscopic residual disease. Ionising radiation (IR) generates a variety of biological changes within irradiated tissue, with double-stranded DNA (dsDNA) breaks being therapeutically significant. IR-induced cell death is also immunogenic, with recruitment of immune effectors to the irradiated tumour microenvironment contributing to a successful outcome.[4] Platinum-based chemotherapy accentuates IR-induced cell death,

in part via the suppression of homologous recombination, the primary repair mechanism for dsDNA damage in irradiated cells.[4] Manipulation of the DDR pathway, therefore, represents a rational means by which the therapeutic index of POCRT might be improved. Cell cycle checkpoints are integral components of the DDR, allowing the cell to pause progression through the cell cycle to repair DNA damage. Checkpoints are regulated by a network of phosphorylation cascades. *p53*, encoded by *TP53*, is a key regulator of the $G_1$/S checkpoint. *TP53* mutations, which are seen in 60%–70% of HNSCC cases,[5] are sufficient to impair the function of this checkpoint and thereby create a critical reliance on the later $G_2$/M checkpoint. In addition, p53 function can be inactivated by various mechanisms, including somatic and germline mutations as well as polymorphisms.[6 7]

Pharmacological abrogation of the $G_2$/M checkpoint has been shown to differentially sensitise normal and tumour cells to the effect of DNA damaging agents such as cisplatin and IR.[8]

## WEE1 kinase and AZD1775

WEE1 kinase is a key regulator of the $G_2$/M checkpoint and a promising therapeutic target. It is a serine-threonine kinase involved in phosphorylation and inactivation of cyclin-dependent kinase 1 (CDK1), the only 1 of 14 similar proteins to be indispensable for mitotic entry. Along with a number of other proteins, WEE1 triggers $G_2$/M arrest in response to DNA damage. However, inhibition of WEE1 leads to high CDK1 activity, allowing cells to progress through the $G_2$/M checkpoint without the opportunity to repair damaged DNA, leading to potentially catastrophic levels of unrepaired DNA damage induction of cell death.[9 10] WEE1 also has an effect on CDK2 as its inhibition leads to high CDK2 activity and aberrant DNA replication, resulting in stalled replication forks and DNA double-stranded breaks. WEE1 upregulation is seen in a variety of human cancers and is inversely associated with prognosis in some models.[11 12] Two separate kinomic screens in HNSCC identified WEE1 expression as a particularly strong determinant of cell survival,[13 14] indicating that HNSCC may be a fruitful setting in which to investigate the clinical effects of WEE1 inhibition.

Adavosertib (AZD1775) is a potent, selective small molecule inhibitor of WEE1. It has been shown to potentiate the activity of various chemotherapeutic agents in vitro and in vivo. Some studies suggest the sensitising effect is only seen in p53-deficient tumours[14–16] although not exclusively.[10] It has also been shown to enhance IR-induced cell death in TP53-mutant cell lines.

Coexposure of AZD1775 and cisplatin were found to reduce clonogenic survival,[17] demonstrating this combination therapy has the ability to overcome cisplatin resistance in HNSCC. Similar effects of this compound on radiation-induced cell death have been seen in models of typically radio-resistant cancer, such as pontine glioma,[18] glioblastoma[19] and pancreatic adenocarcinoma.[20] Importantly, one study has shown that WEE1 inhibition by

AZD1775 sensitises acute myelogenous leukaemia and lung cancer cell lines to cytarabine chemotherapy independently of p53 status,[10] suggesting that p53 mutation as a predictive biomarker for response to WEE1 inhibition may be cancer and/or chemotherapy specific. WEE1 has also been implicated in maintaining genomic stability through stabilisation of replication forks—downregulation reduces replication fork speed during S-phase, generating potentially lethal dsDNA breaks.[21] By impacting both cell cycle progression and DNA damage repair, WEE1 inhibition may potentiate cell death in response to chemotherapy and IR. This suggests that there may be an additive effect on clinical outcome in combination with POCRT, as well as potential synergy.

AZD1775 is being tested in many clinical settings including in combination with docetaxel and cisplatin in HNSCC (NCT02508246),[22] with radiotherapy in childhood pontine glioma (NCT01922076), with temozolomide and radiotherapy in glioblastoma (NCT01849146), and with cisplatin and radiotherapy in cervical cancer (NCT01958658).

In summary, the available mechanistic data lend strong support to combining AZD1775 with cisplatin and with POCRT in the clinic. Given that the predictive effect of TP53 mutation on such combinations has yet to be clinically validated, this study does not prospectively stratify patients based on any such clinical or biological characteristic. Biomarker data generated, however, will be important in understanding the mechanism of action and informing future combination studies.

### Incorporating AZD1775 into clinical management

A recent randomised phase III study of neoadjuvant chemotherapy in a Chinese population with locally advanced resectable oral cavity cancer showed that docetaxel, cisplatin and 5-fluorouracil (TPF) treatment generated a response rate of 80%, with an exploratory analysis showing that clinical responders had improved overall survival and local control.[23] There was no improvement in overall survival for the study group as a whole compared with the control group without upfront TPF chemotherapy, possibly because patients who would ordinarily have received POCRT received PORT alone in this study. In an earlier study, TP53 mutation conferred increased resistance to preoperative cisplatin-based chemotherapy in oral cancer.[24] While neoadjuvant chemotherapy has yet to find a definitive role prior to surgery, these findings raise the possibility that chemosensitisation may further improve outcomes.

The preoperative group A sets out to test this hypothesis by combining 1 week of AZD1775 alone, with a second week of cisplatin (40 mg/m$^2$) with AZD1775. This short course of treatment will minimise any delay between diagnosis and definitive surgery and prevent unacceptable delay for the non-responders from getting standard treatment. If this is a promising combination, the long-term aim is to test AZD1775 in the neoadjuvant setting to downstage the disease through treatment with

cisplatin and AZD1775 in the preoperative setting, which may decrease the extent of surgery, and may also reduce the need for PORT/POCRT by reducing the involved margin rate. Many patients with tumours that have high-risk histopathological features develop relapsed disease within the anatomical area treated to a high radiation dose. Chemosensitisation and radiosensitisation via inhibition of the $G_2/M$ checkpoint by AZD1775 has the potential to improve outcomes with acceptable toxicity. This possibility is the basis of the treatment delivered in the group B cohort.

## METHODS

### Wisteria trial objectives

Primary objectives are to determine the recommended dose and safety profile of AZD1775 with cisplatin in the group A preoperative (window-of-opportunity) setting, and with cisplatin/radiotherapy in the group B postoperative setting. Other objectives: obtain information regarding group A and B disease-free survivals; evaluate AZD1775 pharmacodynamic (PD) effects when administered with cisplatin (group A); identify and correlate potential predictive biomarkers with PD markers of DNA damage (group A); determine the rate of R0, R1 or R2 resection status and surgical complication (group A); and group B quality of life (QoL).

### Trial design

Wisteria is an open-label, dose-finding, multicentre phase I trial to determine the highest safest dose of AZD1775 in combination with a single dose of cisplatin in the preoperative setting as a window of opportunity trial (group A); and to determine the maximum tolerated dose (MTD) of AZD1775 in combination with cisplatin/radiotherapy in the postoperative setting (group B): both groups recruit and run in parallel (figure 1). The dose-escalation design is based on the modified time-to-event continual reassessment method (TITE-CRM),[25–28] requiring a maximum of 21 patients per group, and encompassing up to four dose levels of AZD1775.

The MTDs of AZD1775 for both groups are expected to differ given the additional toxicities of radiotherapy in group B. The dose-limiting toxicity (DLT) monitoring periods are defined in table 1. Conservative target DLT rates are selected to minimise the likelihood of compromising individual patient's radical surgery and/or PORT. To maximise recruitment, reduce trial suspension time between cohorts, while balancing safety and optimal patient allocation, screening cohorts of up to five patients will be allowed if the dose has previously been tested. Recruited patients will be allocated to the current recommended dose up to a maximum of five. Patients who become unevaluable may be replaced.

The model will be updated after every two to three evaluable patients, with any subsequent eligible patients (not already receiving treatment) allocated to the latest recommended dose cohort. Subsequent cohorts will be

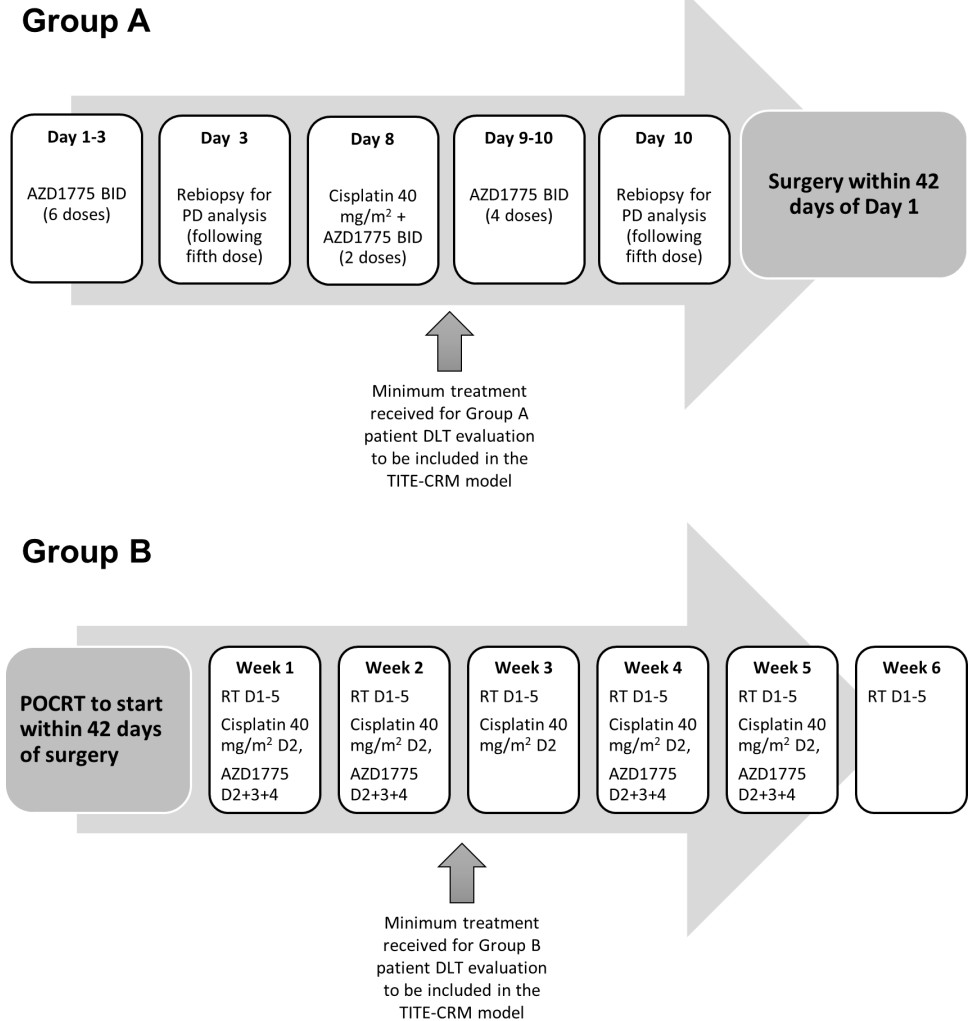

**Figure 1** Wisteria trial schema. Group A patients were required to have received a minimum treatment of chemotherapy (cisplatin) and AZD1175 in combination to be DLT evaluable. Group B patients were required to have received a minimum of 2 weeks of treatment (half the total scheduled AZD1775 dose) to be DLT evaluable. DLT, dose-limiting toxicity; PD, pharmacodynamic; TITE-CRM, time-to-event continual reassessment method.

assigned a dose level using all the data observed until either—the MTD is determined, the maximum sample size is reached, or the trial is stopped early due to unacceptable DLT levels at the lowest dose.

### Patient population, screening and consent

Group A patients are those with biopsy-proven squamous cell carcinoma of the oral cavity, larynx or hypopharynx (having accessible tumours for rebiopsy under local anaesthetic or via ultrasound-guided biopsy), and due to undergo surgery. Patients in group B consist of those already diagnosed with oral cavity, larynx or hypopharynx squamous cell carcinoma, who have undergone surgery and are considered at risk of relapse after surgery (ie, with positive margins and/or nodal extracapsular spread). Patients with cancer of the oropharynx will not be included. Since Human papillomavirus (HPV) is only routinely tested in oropharyngeal squamous cell carcinoma or unknown primary of squamous cell carcinoma in the National Health Service (NHS), HPV status was not determined or requested for the tumours as part of

the study. Patients who meet the criteria will be supplied with the patient information sheet. If informed consent (online supplementary appendices 1 and 2) is given, the patient undergoes a full screening evaluation to ensure they satisfy the eligibility criteria prior to registration (boxes 1 and 2).

### Dose selection

Based on AstraZeneca studies,[29 30] the Wisteria trial uses 3-day dosing, giving AZD1775 weekly to coincide with cisplatin administration thereby potentiating the DNA damage effects. The maximum total dose of AZD1775 1800 mg in combination with 1 dose of 40 mg/m$^2$ cisplatin for group A is substantially lower than AZD1775 monotherapy dose 2250 mg every 21 days (225 mg two times per day orally over 2.5 days per week for 2 weeks per 21 day cycle) used in Do *et al*; and although in combination with cisplatin, the highest dose level (dose level 2) is likely to be safe.

A similar dose escalation algorithm will be carried out in group B with a higher acceptable target DLT level of

**Table 1** Details of the Wisteria TITE-CRM trial design for groups A and B

| | Group A | | | Group B | | |
|---|---|---|---|---|---|---|
| Maximum no of Patients | Up to 21 patients in each group | | | | | |
| MTD definition | Dose closest to where 25% patients experience a DLT | | | Dose closest to where 30% patients experience a DLT | | |
| Minimum treatment to be DLT evaluable | Patients to have received the minimum of AZD1775 and cisplatin doses scheduled up to and including day 8 | | | Patients to have received a minimum of the first 2 weeks of treatment as scheduled | | |
| DLT reporting period | Minimum of 30 days from start of treatment up to 42 days for delays in surgery due to treatment related toxicity (if this exceeds 42 days then this delay will be classified as a DLT). | | | Minimum of 56 days (8 weeks) from start of radiotherapy, and up to 84 days (12 weeks), that is, up to 6 weeks from end of postoperative chemoradiation therapy. | | |
| Dose levels | Cisplatin (intravenous) over 1 hour (day 8) | AZD1775 PO two times per day 3 days (days 1–3, 8–10) | Cisplatin (intravenous) over 1 hour (day 2 weeks 1–5) | Radiotherapy (5 days per week over 6 weeks) | AZD1775 PO two times per day 3 days (days 2–4 weeks 1, 2, 4 and 5) |
| -1 | 40 mg/m$^2$ | 75 mg | 40 mg/m$^2$ | 54–65 Gy in 30# | 50 mg |
| 0 (starting dose level) | | 100 mg | | | 75 mg |
| 1 | | 125 mg | | | 100 mg |
| 2 | | 150 mg | | | 125 mg |

DLT, dose-limiting toxicity; MDT, multidisciplinary team; TITE-CRM, time-to-event continual reassessment method.

30% allowing for multiple dosing of cisplatin with radiotherapy in combination with AZD1775.[31–33] Trial doses and schedule are set out in table 1. AZD1775 is administered orally. The feasibility of the oral administration will be evaluated in postoperative patients.

## Trial treatments

### Group A (preoperative)
Patients will receive specified dose of AZD1775 by mouth (PO) two times per day at 12 hours intervals (2 hours before or after food) for 3 days, commencing on days 1 and 8, with 40 mg/m$^2$ intravenous cisplatin delivered over 1 hour on day 8. A missed dose of more than 6 hours should be skipped. If vomiting occurs, patient should not retake the dose but wait until next scheduled dose. Surgery is not to exceed 42 days from start of neoadjuvant chemotherapy.

### Group B (postoperative)
Patients will receive specified dose of AZD1775 PO two times per day for 3 days, commencing on days 2, 9, 23 and 30, and 40 mg/m$^2$ intravenous cisplatin delivered over 1 hour on days 2, 9, 16, 23 and 30 (days are timed from the start of radiotherapy delivery). Radiotherapy (65 Gy and 54 Gy in 30 fractions given for 5 days per week over 6 weeks: D1-5, D8-12, D15-19, D22-26, D29-33, D36-40, to high-risk and low-risk volumes, respectively, as per Wisteria Radiotherapy Guidelines, see online supplementary appendices 3–6) will commence within 3 months of surgery and given concurrently with chemotherapy.

For both groups A and B, cisplatin dose banding is not permitted. BSA to be calculated according to local practice; if weight changes>10% then cisplatin dose should be recalculated.

AZD1775 (morning dose) should be administered prior to cisplatin, and group B to receive cisplatin 1 hour before radiotherapy; standard supportive treatment, including premedication with antiemetics (excluding aprepitant and fosaprepitant), is allowed according to standard practice guidelines.

## Trial outcome measures

### Primary outcome measures
The recommended doses of AZD1775 (determined by the modified TITE-CRM) for group A and B.
► Group A: the highest safe dose in combination with cisplatin with a predefined target DLT probability of 25% for up to 42 days from start of treatment.
► Group B: the MTD in combination with cisplatin/radiotherapy with a target DLT of 30% for up to 12 weeks from the start of treatment.
► Safety profile of AZD1775 in both group A and B.

### Secondary outcome measure
Disease-free survival time for both groups and recorded from commencement of treatment to the date of disease recurrence, patient death or end of trial follow-up period.

### Tertiary research outcome measures
PD effects of AZD1775 and correlation with TP53 mutation status; pharmacokinetic (PK) effects of AZD1775 will

## Box 1 Inclusion criteria—all patients

- ► Histologically confirmed diagnosis of oral, laryngeal or hypopharyngeal squamous cell carcinoma.
- ► Multidisciplinary team (MDT) recommendation for surgical resection with curative intent.
- ► Eastern Cooperative Oncology Group performance status 0/1[31].
- ► Age≥18 to≤70 years.
- ► Creatinine clearance, measured by glomerular filtration rate (GFR),≥60 mL/min at baseline calculated using local practice calculation. If this is ≤60 mL/min then an isotopic GFR may be carried out and must be >60 mL/min.
- ► Acceptable cardiac function. If significant cardiac history, then required for patient to have left ventricular ejection fraction ≥55% by echocardiogram (ECHO) or multiple gated acquisition scan (if ECHO is equivocal).
- ► Normal liver and bone marrow function:
  - – Haemoglobin ≥100 g/L.
  - – Absolute neutrophil count ≥1.5×10$^9$/L.
  - – Absolute platelet count ≥100×10$^9$/L.
  - – Aspartate transaminase or alanine aminotransferase ≤2.5 upper limit of normal (ULN).
  - – Total bilirubin ≤1.5 ULN (except for patients with known Gilbert's syndrome).
- ► Male and female participants must agree to take appropriate measures to prevent pregnancy. Contraceptive measures should be used for 2 weeks prior to trial entry, during the trial and for at least 6 months after last receiving treatment. Acceptable methods of contraception include total abstinence (if this is the patient's usual and preferred lifestyle choice), tubal ligation, combined oral, transdermal or intravaginal hormonal contraceptives, medroxyprogesterone injections (eg, Depo-Provera), copper-banded intrauterine devices; hormone impregnated intrauterine systems and vasectomised partners. All methods of contraception (with the exception of total abstinence) should be used in combination with the use of a condom by their male sexual partner for intercourse.

In addition to general criteria
- ► Group A—Accessible tumours for rebiopsy to be taken under local anaesthetic or via ultrasound-guided biopsy.
- ► Group B—High-risk histopathological features after surgical resection, that is, nodal extra-capsular spread and/or tissue resection margin <1 mm as agreed at MDT.
- ► Group B—Patients who have previously registered to group A can be considered for inclusion in group B.

## Box 2 Exclusion criteria—all patients

- ► Any previous treatment for the same cancer, or previous head and neck malignancy, apart from laser excision of carcinoma in situ, with minimal residual functional deficit or registration and treatment in group A prior to surgery.
- ► Patients with cancer of the oropharynx or non-primary cancer will not be included.
- ► Any metastatic disease from any primary site.
- ► Use of an investigational medicinal product concurrently or within 4 weeks of starting this trial.
- ► Uncontrolled intercurrent illness, which will interfere with patient's trial participation, for example,
  - – Myocardial infarction within 6 months.
  - – Congestive cardiac failure.
  - – Unstable angina.
  - – Symptomatic cardiomyopathy.
  - – Chronic infections.
  - – Active peptic ulcer or liver disease.
  - – Serious psychiatric condition limiting ability to comply with trial protocol.
- ► Clinical evidence of current heart failure (≥New York Heart Association Class II[32]).
- ► Clinical evidence of atrial fibrillation (heart rate >100 bpm, within 6 months prior to trial entry).
- ► Unstable ischaemic heart disease (myocardial Infarction within 6 months prior to trial entry or angina requiring the use of nitrates greater than once weekly).
- ► Patients who have a history of torsades de pointes (unless all risk factors that contributed to torsades have been corrected).
- ► Active gastrointestinal disease that might limit absorption of study drug, for example, coeliac disease, Crohn's disease, ulcerative colitis, pancreatic insufficiency.
- ► Evidence of any psychological, familial, sociological or geographical condition potentially hampering protocol compliance
- ► Participation in another interventional clinical trial while taking part in this trial.
- ► Patients who are unable to discontinue any prohibited drug, including live vaccines and unable to tolerate a washout period for at least 14 days prior to trial entry (including: CYP3A4 inhibitors; CYP3A4 inducers; CYP3A, CYP3A4, CYP2C19, CYP1A2 sensitive substrates or substrates with narrow therapeutic range; P-gp substrates, strong P-gp inhibitors and BCRP substrates[33]).
- ► Patients with any contraindications to cisplatin use.
- ► Clinical judgement by the investigator that the patient should not participate in the study.
- ► Known hypersensitivity to the study drugs or active substances or excipients of the preparations.
- ► Pregnant or breastfeeding patients.
- ► Significant pre-existing neuropathy which currently interferes with the patient's daily life.
- ► Mean resting corrected QTc interval using the Fridericia's formula[34] >450 ms (male) and >470 ms (female) (as calculated per institutional standards) obtained from three ECGs 2–5 min apart at study entry, or congenital long QT syndrome.
- ► Inability to swallow oral medications.

be determined; optimise, validate and test feasibility of assays to investigate serum, ctDNA and RNA biomarkers; investigate the feasibility of immune function testing in a multicentre setting; complete pathological response rate for group A; positive resection margin status in group A; surgical complication in group A; and QoL in group B using European Organization for Research and Treatment of Cancer (EORTC) C30 (V.3.0),[34] EORTC QLQ-H&N35[35] and M.D. Anderson Dysphagia Inventory,[36] as summarised in table 2.

Assessment data will be collected as listed in online supplementary appendix tables 7 and 8.

### DLT and dose management

Predefined DLTs have been specified by the Clinical Investigators (box 3), and recorded toxicity profiles

for the Wisteria treatments are given in online supplementary appendix table 9. Toxicity is assessed using the National Cancer Institute Common Terminology Criteria for Adverse Events V.4.0[37] unless otherwise specified. Any

**Table 2**  Trial summaries of group A and group B

| Summary of group A | |
| --- | --- |
| Setting | Patients undergoing surgical resection |
| Design | Modified TITE-CRM |
| Chemotherapy | Cisplatin 40 mg/m$^2$ intravenous over 1 hour on day 8 |
| AZD1775 | AZD1775 PO BID for 3 days on days 1–3 and 8–10 dose-recommendation according to the modified TITE-CRM model |
| Surgery | Resection within 42 days of start of neoadjuvant treatment |
| DLT reporting period | The minimal reporting period is 30 days from start of treatment, but patients will be monitored up to 42 days for delay in surgery due to treatment-related toxicity |
| PK samples | Pharmacokinetic samples will be collected pre and post—the fifth dose of AZD1775 on days 3 and 10 (4 samples per patient) |
| PD markers | Assess CDK1, pCDK1, γH2AX, p53, p16, HH3, pHH3, Ki67, C3, CC3, p21, WEE1, pWEE1, PDL-1 and TILS, and other markers of particular interest |
| Follow-up | Clinically for at least 12 weeks from start of treatment |
| Summary of group B | |
| Setting | Postoperative patients with high-risk disease (involved resection margins +/or extracapsular nodal spread) receiving chemoradiation |
| Design | Modified TITE-CRM |
| Chemotherapy | Cisplatin 40 mg/m$^2$ intravenous over 1 hour on day 2 of weeks 1–5 of radiotherapy |
| Radiotherapy | External beam radiation therapy (30 fractions over 6 weeks) to start within 3 months of surgery. Dose levels are 65 Gy for positive margin, 60 Gy to lymph node levels that have been dissected and 54 Gy to elective lymph node areas. A dose of 65 Gy may also be applied to areas of gross extracapsular spread at the discretion of the treating clinician. |
| AZD1775 | AZD1775 PO BID for 3 days on days 2–4 of weeks 1, 2, 4 and 5 (no treatment with AZD1775 during weeks 3 and 6). Dose recommendation will be according to modified TITE-CRM model. |
| DLT reporting period | The minimal reporting period is 56 days (8 weeks) from start of radiotherapy, but patients will be monitored for DLTs up to 84 days (12 weeks), that is, up to 6 weeks from the end of POCRT |
| PK samples | Pharmacokinetic samples will be collected pre and post—the fifth dose of AZD1775 on week 1—day 4 (2 samples per patient) |

DLT, dose-limiting toxicity; PD, pharmacodynamic; PK, pharmacokinetic; POCRT, postoperative chemoradiation; TITE-CRM, time-to-event continual reassessment method.

patient requiring a toxicity-related dose delay of more than 21 days must be discontinued from the study unless there is approval from the chief investigator for the patient to continue. For group A, if any treatment-related grade≥3 toxicity develops in week 1, cisplatin and AZD1775 are discontinued in week 2. For group B, if calculated GFR falls below 60 mL/min, immediately before any cycle of concomitant chemotherapy, weekly cisplatin should be discontinued and consideration given to substitute with weekly carboplatin (Area Under the Curve (AUC)=1.5), according to routine local practice.

In both groups, a full blood count is obtained at the beginning of each week. If haematological toxicity occurs, treatment should be modified as per online supplementary appendix tables 10 and 11. For non-haematological toxicities, treatment should also be modified as per online supplementary appendix tables 12–16.

The selection of PD markers to test are different for group A and group B as group A enables examination of samples pretreatment and post-treatment with AZD1775.

### Concomitant medications

All concomitant medications received within 14 days before the first dose of study medication and for 12 weeks after the last dose of study medication should be recorded. Medications may be administered for maintenance of existing conditions prior to study enrolment or for a new condition that develops while on study. The treatments and medications listed in online supplementary appendix 17 are prohibited or to be used with caution while in this study. No other investigational therapy or anticancer agents, other than the study medications, should be given to patients. If such agents are required for a patient, then the patient must first be withdrawn from the study. Live vaccines are not permitted.

### Discontinuation of investigational medicinal product

Patients should discontinue trial treatment in the following circumstances: intolerable toxicity; confirmed disease recurrence; pregnancy; severe non-compliance to protocol; development of any study-specific criteria for

## Box 3 Dose-limiting toxicities (DLT)

Patients experiencing any of the following adverse reactions (ARs) during the reporting period will be considered to have experienced a DLT:

► Grade 3 or 4 neutropenia lasting for >7 days despite adequate granulocyte-colony stimulating factor (G-CSF) support.
► Grade 4 febrile neutropenia, which includes grade 3 febrile neutropenia accompanied by systemic inflammatory response syndrome/sepsis or other life-threatening consequences.
► A third occurrence of grade 2 or worse neutropenia despite G-CSF support.
► Grade 3 or 4 thrombocytopenia lasting ≥7 days.
► Any occurrence of grade 4 anaemia.
► Any occurrence of grade 4 mucositis.
► Any occurrence of grade 3 nausea despite preventative and supportive care according to local practice. Tube feeding is not considered a DLT.
► Any occurrence of grade 4 vomiting despite preventative and supportive care according to local practice. Tube feeding is not considered a DLT.
► Any occurrence of grade 4 diarrhoea despite preventative and supportive care according to local practice.
► Any AR which results in an omission of chemotherapy administration and/or study treatment for >14 days.
► A start-of-treatment-to-surgery time of >42 days in group A as a result of a treatment-related toxicity.
► Grade 3 mucositis lasting >42 days after the end of treatment in group B.
► An overall treatment time of >49 days from the first day of radiotherapy in group B as a result of a treatment-related toxicity.
► Any grade 3 or 4 non-haematological AR (except for fatigue, nausea, vomiting and diarrhoea) for which medical intervention that lasts >7 days is required and the investigators deem that this AR is more severe or prolonged than what would be expected of standard treatment. Tube feeding is not considered a DLT.
► Any clinically significant occurrence which investigators within the trial management group agree would place the patient at undue safety risk

If the inability to swallow the AZD1775 capsule develops during the course of trial treatment, this event should be noted in the relevant section on the Suspected DLT form and reported immediately. However, this is not considered a DLT in itself.

## Box 4 Adverse event and serious adverse event definitions

### Adverse event
Any untoward medical occurrence in a patient or clinical trial subject administered a medicinal product and which does not necessarily have a causal relationship with this treatment.

### Serious adverse event
Any untoward medical occurrence or effect that at any dose may:
► Result in death.
► Is life threatening.
► Requires hospitalisation or prolongation of existing in-patient hospitalisation.
► Results in persistent or significant disability or incapacity.
► Is a congenital anomaly/birth defect.
► Is considered medically significant by the clinical investigator.

discontinuation and investigator decision, for example, if the patient requires a prohibited concomitant medication.

### Safety monitoring: adverse events and serious adverse events

All medical occurrences which meet the definition of an adverse event (AE) or serious AE (SAE) (as defined in box 4) should be reported. This includes abnormal laboratory findings of grade 3 and above only. AEs will be monitored for and reported from date of informed consent until the 12-week follow-up visit for group A and until the 12-month follow-up visit for group B. Investigators should report SAEs within 24 hours of first knowledge of the event, until 12-week follow-up visit. After this time, expedited reporting is no longer required and should be reported as an AE. Any poststudy serious unexpected serious adverse reactions (SUSARs) should be reported

within 24 hours. An independent Safety Committee will review all SAEs. Fatal or life-threatening SUSARs will be reported to the Medicines and Healthcare products Regulatory Agency (MHRA) and Research Ethics Committee (REC) within 7 days. Detailed follow-up information will be provided within an additional 8 days. All other events categorised as SUSARs will be reported within 15 days. The MHRA and REC will be notified immediately if a significant safety issue is identified during the course of the trial. All SAEs relating to AZD1775 will be reported to AstraZeneca within 24 hours of notification.

### Data collection and monitoring

Data will be collected via a set of forms capturing details of eligibility, baseline characteristics, treatment and outcome details. This trial will use an electronic remote data capture system. SAE reporting and notification of pregnancy will be paper based. All missing and ambiguous data will be queried. In all cases, it remains the responsibility of the investigator to ensure that data are accurate. Details regarding data collection for group A and B are given in online supplementary appendix tables 7 and 8. The Investigator will permit trial-related monitoring, audits, ethical review and regulatory inspection(s) at their site, providing direct access to source data/documents.

### Statistical methodology

The AZD1775 dose is allocated to patients according to a modified Bayesian TITE-CRM, using an empiric dose-toxicity model, F(x,β) given as:

$$F(x, \beta) = x^{\exp(\beta)} \ for \ 0 < x < 1$$

where model parameter β is assumed random and follows a normal distribution and will be estimated by its posterior mean.[27] At the point of model update, patients who have started treatment but have not experienced DLT (and have completed their full DLT assessment) will be included in the probability calculation with a weight equal to the proportion of the full DLT assessment period they have completed. Patients who experience

DLT within the assessment period or complete the full assessment period are assigned full weight. The model also enables inclusion of partial patient information that cannot be formally evaluated for DLTs due to withdrawal, treatment discontinuation or death, which are unrelated to treatment, within the DLT assessment period.

Table 1 displays the four dose levels of AZD1775 for group A and group B. For both groups, dosing starts at level 0 and allows for possible escalation to two higher levels, or deescalation to a lower dose, as recommended by the TITE-CRM, without skipping untried doses in escalation. The TITE-CRM model aims to recommend the next dose with estimated DLT probability closest to the target DLT level, taking into account the practical considerations as detailed below:

### Practical considerations
► Restriction is applied to avoid skipping of untried doses in escalation.
► Stop early due to safety concerns if there is sufficient evidence that the posterior probability of DLT at the lowest dose is greater than the target DLT rate, implying that the lowest dose is too toxic.
► Stop early if sufficient patients have been allocated to the current MTD (and remains the recommended dose level for the next cohort if the trial continues) before the full recruitment of 21 patients.
► The minimum DLT period is set at 30 days for group A and 8 weeks for group B. This means that partial information of no DLT in a patient can only be included in the model if they have been followed up for at least 30 days or 8 weeks in group A and B, respectively. (The full DLT assessment periods are 42 days and 12 weeks in group A and B, respectively.) This feature can easily be accommodated using the TITE-CRM model.
► A 'look ahead' strategy will be implemented if the next recommended dose level by the modified TITE-CRM model will not be influenced by the outcome of the remaining patient(s) of a particular cohort (DLT or no DLT). By implementing this strategy, the next cohort can be recruited immediately without awaiting the final observations from the current cohort, thus reducing waiting time between cohorts.

### Analysis of outcome measures
The primary outcome measures are the recommended doses of AZD1775 in group A and group B, and the safety profile of the combination therapy. To be DLT evaluable, group A patients must have received AZD1775 and cisplatin doses scheduled up to and including day 8; group B patients must have received at least the first 2 weeks of treatment. The recommended doses of AZD1775 will be those that have an estimated DLT rate closest to the target DLT rate: 25% for group A; 30% for group B. DLT rates and corresponding 90% probability intervals will be reported. Safety will be monitored in patients treated with at least one dose of the trial treatment until end of follow-up. All adverse reactions, AEs, SAEs, suspected

unexpected SUSARs, deaths, deviations and withdrawals experienced by trial patients, throughout the trial's duration, will be reported separately for group A and group B.

Disease-free survival in group A and group B patients will be reported separately. Patients will be followed up for 12 months after which, time-to-event outcomes will be assessed using the method of Kaplan and Meier. Median disease-free survival with corresponding 95% CIs will be reported where appropriate. Further analysis will be performed separately on subsets of group A and B patients who have completed at least 90% of AZD1775 trial scheduled treatments.

Tertiary outcome measures will be explored and reported descriptively using basic descriptive statistics where appropriate:
► PD effects of AZD1775 and correlation with TP53 mutation status: Expression levels of parameterschanges over time and correlation with TP53 mutation status will be reported.
► PK effects of AZD1775: Blood samples for PK analyses are collected (as scheduled in online supplementary appendix tables 7 and 8). Circulatory levels of AZD1775 and changes over time will be reported.
► Complete pathological response rate for group A: Pathology data will be reported along with the results of the formalin-fixed paraffin-embedded (FFPE) tissue samples. Response rate will be reported as the number of evaluable group A patients showing a response (numerator) divided by the total number of group A evaluable patients (denominator).
► Positive resection margin status in group A: This will be reported as the number and percentage of group A patients showing positive resection margins.
► Surgical complications in group A: This will be reported by category of surgical complication with severity and given as numbers and percentages.
► QoL in group B: QoL data are gathered over time and will be reported as changes over time and plotted. In addition to the descriptive analyses, the repeated measures over time data may be modelled, if appropriate, with a linear mixed effects model (taking subject correlation into account) using linear or more flexible models.

Interim analyses will be performed once each cohort of patients has been recruited and assessed within the defined assessment timeframe of DLTs. In each meeting, the safety committee will be presented with recruitment and safety data, together with a statistical report for next dose recommendation. Additional meetings might be convened if late-onset DLTs were observed. The safety committee will decide whether to progress to the recommended dose as indicated by the modified TITE-CRM model. The final analyses will be conducted in two parts: (1) analyses of the primary outcome measures of MTDs for groups A and B, to be carried out approximately 3 months after the safety review of the last cohort in each group and (2) longer-term outcomes—including secondary and tertiary outcome measures and updated safety data, to be carried out 1 year after the end of trial.

## TRIAL ORGANISATION AND STRUCTURE

The trial is sponsored by the University of Birmingham, UK and conducted under the auspices of the Cancer Research UK Clinical Trials Unit, University of Birmingham, UK according to their local procedures.

The trial management group (TMG) will include as a minimum: chief investigator, coinvestigators, lead and trial statisticians, trial management team leader and trial coordinator (other appropriate key personnel will be invited to attend meetings as required). This group will be responsible for legal obligations, day-to-day running and management of the trial.

The safety committee provides support on decisions surrounding the review of DLT information and dose-changing decisions. Membership of this group includes the TMG, independent members and selected principal investigators.

## ETHICS AND DISSEMINATION

The trial will be performed in accordance with the 18th World Medical Association General Assembly, Helsinki, Finland, June 1964, amended at the 48th World Medical Association General Assembly, Somerset West, Republic of South Africa, October 1996 (website: http://www.wma.net/en/30publications/10policies/b3/index.html); conducted in accordance with the Research Governance Framework for Health and Social Care, the applicable UK Statutory Instruments, (which include the Medicines for Human Use Clinical Trials 2004 and subsequent amendments, The Human Tissue Act 2008) and Good Clinical Practice E6(R2); and carried out under a Clinical Trial Authorisation in accordance with the Medicines for Human Use Clinical Trials regulations. The protocol and subsequent amendments will be submitted to and approved by REC prior to circulation. Personal data recorded on all documents will be regarded as strictly confidential and will be handled and stored in accordance with the General Data Protection Regulation 2016/679 and the Data Protection Act (2018).

Trial findings will be published in a peer-reviewed journal and disseminated at appropriate conferences, departmental and scientific meetings.

**Author affiliations**
[1]Institute of Cancer and Genomic Sciences, University of Birmingham, Birmingham, UK
[2]University Hospitals Birmingham NHS Foundation Trust, Birmingham, UK
[3]Cancer Research UK Clinical Trials Unit, Institute of Cancer and Genomic Sciences, University of Birmingham, Birmingham, UK
[4]SIMBEC Research Limited, Merthyr Tydfil, UK
[5]University College London, London, UK
[6]Beatson West of Scotland Cancer Centre, University of Glasgow, Glasgow, UK
[7]Institute of Cancer Research, London, UK
[8]Clatterbridge Cancer Centre NHS Foundation Trust, Bebington, UK
[9]Leeds Teaching Hospitals NHS Trust, Leeds, UK
[10]Institute of Immunology and Immunotherapy, University of Birmingham, Birmingham, UK

**Acknowledgements** Wisteria was supported by Experimental Cancer Medicine Centres (ECMC) funding and by the ECMC Network.

**Contributors** The study was conceived by HM and JG; designed by HM, JG, AKo and CY; and the protocol written by HM, AKo, JG, CY, LL, RM, AKi, RS, JSav, JSac, MF, JP, SS, PM, GM and KH. AKi, AKo, JG, JSav and CY wrote the manuscript with input from all authors.

**Funding** Cancer Research UK (code C19677/A20959) and AstraZeneca through the CRUK's Combinations Alliance and Experimental Cancer Medicine Centre (ECMC). AstraZeneca provides AZD1775 to participating sites free-of-charge, and was consulted over the trial design but are not involved in the trial management group or safety committee. The Combinations Alliance, a Cancer Research UK model, aims to drive academic-industrial partnership, generating novel treatment options that would otherwise unlikely be realised. Novel combination ideas are generated and delivered via the UK's Experimental Cancer Medicine Centre (ECMC) network of clinical and scientific experts working together to accelerate innovation in early-phase oncology research for patient benefit.

**Competing interests** None declared.

**Patient consent for publication** Not required.

**Provenance and peer review** Not commissioned; externally peer reviewed.

**ORCID iDs**
Joshua Savage http://orcid.org/0000-0003-0599-0245
Christina Yap http://orcid.org/0000-0002-6715-2514
Hisham Mehanna http://orcid.org/0000-0002-5544-6224

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
