## [Reviewer comments · BMJ Open]

ARTICLE DETAILS

TITLE (PROVISIONAL)	A Phase I trial of WEE1 inhibition with Chemotherapy and Radiotherapy as adjuvant treatment, and a Window of Opportunity trial with Cisplatin in Patients with Head and Neck Cancer: The WISTERIA Trial Protocol
AUTHORS	Kong, Anthony; Good, James; Kirkham, Amanda; Savage, Joshua; Mant, Rhys; Llewellyn, Laura; Parish, Joanna; Spruce, Rachel; Forster, Martin; Schipani, Stefano; Harrington, Kevin; Sacco, Joseph; Murray, Patrick; Middleton, Gary; Yap, Christina; Mehanna, H

VERSION 1 - REVIEW

REVIEWER	Ruud Brakenhoff Amsterdam UMC, Otolaryngology/head and neck surgery, the Netherlands
REVIEW RETURNED	21-Aug-2019

GENERAL COMMENTS	Kong et al present the clinical protocol of the Wisteria trial. In group A neoadjuvant AZD1775 is given with a single dose of cisplatin and in group B adjuvant AZD1775 in surgically treated patients with high-risk features for relapse. Patients of group A can be switched to B. Primary endpoint is assessment of DLT. It is certainly helpful to have these protocols published. The trial is interesting, but I have a few comments nonetheless: 1) HPV is not tested. Given the patient population this is not necessary but the consideration should be discussed.2) Please stick to CDK1 notation throughout and do not use CDC2 which is the yeast name.3) I did not see the citation to Mendez et al CCR 2018 on AZD in HNSCC4) The role of AZD in the trial organization should be mentioned (page 20). Did they provide the drug for free, was their data used to get Ethics approval etc.
---

VERSION 1 – AUTHOR RESPONSE

1) HPV is not tested. Given the patient population this is not necessary but the consideration should be discussed.

A statement has been added to the "Patient population, screening and consent" section to address this.

2) Please stick to CDK1 notation throughout and do not use CDC2 which is the yeast name.

This has been corrected throughout.

3) I did not see the citation to Mendez et al CCR 2018 on AZD in HNSCC.

A citation has been added to introduction when other studies using AZD1775 are discussed.

4) The role of AZD in the trial organization should be mentioned (page 20). Did they provide the drug for free, was their data used to get Ethics approval etc.

Further clarity regarding the role of AstraZeneca has been added to the Funding section.